# The Crosstalk Between Cartilage and Bone in Skeletal Growth

**DOI:** 10.3390/biomedicines12122662

**Published:** 2024-11-21

**Authors:** Frank Hernández-García, Ángela Fernández-Iglesias, Julián Rodríguez Suárez, Helena Gil Peña, José M. López, Rocío Fuente Pérez

**Affiliations:** 1Departamento de Medicina, Oviedo University, 33003 Oviedo, Spain; frankhernandezgarcia1996@gmail.com (F.H.-G.); rodriguezjulian@uniovi.es (J.R.S.); 2Grupo Investigación Pediatría, Instituto de Investigación Sanitaria del Principado de Asturias, 33011 Oviedo, Spain; angelafiglesias@gmail.com (Á.F.-I.); gilpena@gmail.com (H.G.P.); jmlopez@uniovi.es (J.M.L.); 3AGC de Infancia y Adolescencia, Hospital Universitario Central de Asturias, 33011 Oviedo, Spain; 4RICORS-SAMID (RD21/0012), Instituto de Salud Carlos III, 28029 Madrid, Spain; 5RICORS2040 (RD21/0005/0011), Instituto de Salud Carlos III, 28029 Madrid, Spain; 6Departamento de Morfología y Biología Celular, Oviedo University, 33003 Oviedo, Spain; 7Universidad Europea de Madrid, Department of Nursing, Faculty of Medicine, Health and Sports, 28670 Madrid, Spain

**Keywords:** chondrocytes, osteocyte, osteogenesis, growth plate, endochondral ossification, cell–cell interactions

## Abstract

While the flat bones of the face, most of the cranial bones, and the clavicles are formed directly from sheets of undifferentiated mesenchymal cells, most bones in the human body are first formed as cartilage templates. Cartilage is subsequently replaced by bone via a very tightly regulated process termed endochondral ossification, which is led by chondrocytes of the growth plate (GP). This process requires continuous communication between chondrocytes and invading cell populations, including osteoblasts, osteoclasts, and vascular cells. A deeper understanding of these signaling pathways is crucial not only for normal skeletal growth and maturation but also for their potential relevance to pathophysiological processes in bones and joints. Due to limited information on the communication between chondrocytes and other cell types in developing bones, this review examines the current knowledge of how interactions between chondrocytes and bone-forming cells modulate bone growth.

## 1. Introduction

Longitudinal growth is a continuous, multifactorial process that is genetically determined and influenced by multiple factors such as nutrition, environment, and hormones. Normal growth reflects a child’s overall health and proceeds in a predictable fashion: it is rapid during the first two years of life and gradually slows down with age until puberty. During puberty, a second growth spurt occurs, and growth eventually stops when the growth plate closes, marking the achievement of the individual’s final size [1,2,3].

Endochondral ossification in the growth plate (GP) plays an important role in developing bone mineral mass, which enables the elongation of bones into adulthood. This complex process is highly coordinated, involving the action of chondrocytes, osteocytes, osteoblasts, and osteoclasts through intricate signaling pathways among the cells. Effective communication between these cell types ensures the proper development and maintenance of bone structure, making this crosstalk essential for normal skeletal growth.

Disruptions in these communication pathways can lead to growth disorders, with short stature or growth failure being among the most common clinical outcomes. In general, more severe growth retardation increases the likelihood of an underlying pathological condition. Short stature in children is typically defined as a height below the third percentile or at least two standard deviations (SD) below the mean for their age and sex. Short stature may arise idiopathically, result from genetic disorders affecting bone growth, or occur secondary to systemic diseases. It encompasses a range of disorders with varied causes, including malnutrition, hormonal imbalances, environmental stress, or disease states such as chronic kidney disease (CKD) [4,5].

Given the critical role of the GP in skeletal development, understanding the cellular and molecular mechanisms controlling its function is key to identifying potential therapeutic strategies for treating growth-related disorders. In this review, we will explore the current knowledge of the cell–cell interactions and signaling pathways that drive bone growth, with a particular emphasis on the interplay between chondrocytes, bone-forming cells, and surrounding tissues. Particular emphasis will be given to diseases that disrupt these interactions and providing insights into conventional and new potential therapeutic targets for these growth-related disorders.

## 2. Physiology of Bone Growth

In the human skeleton, there are mainly two types of ossification: intramembranous and endochondral ossification. Both processes are highly important in developing different skeletal structures. Intramembranous ossification mainly gives rise to flat bones, such as skull and clavicle bones, via the direct mineralization of mesenchymal tissue. In contrast, endochondral ossification regulates the development of long bones, such as those of limbs, and other structures that have to grow both in length and in resistance to mechanical forces (Figure 1 and Figure 2).

### 2.1. The Growth Plate and Endochondral Ossification

The GP, also called the epiphyseal plate, physis, or growth cartilage (Figure 2), is the organ responsible for longitudinal growth and is the site where new bone formation occurs through endochondral formation. The primary biomechanical role of the GP is to facilitate the lengthening of bones. The GP also helps to absorb mechanical stress and allows bones to grow under various levels of pressure and tension. Because cartilage is more flexible than bone, growth plates help to prevent fractures and injuries in growing bones, which are more susceptible to impact forces. Like bones, growth plates can adapt to certain levels of mechanical loading. However, they are more sensitive than mature bone tissue. Excessive stress or trauma to the growth plate can disrupt its normal function and produce growth arrest.

Bone formation begins in utero around the sixth or seventh week of gestation, starting as a cartilage model template [6]. This cartilage grows on the epiphyseal side and ossifies at the diaphyseal side. In the late teens or early twenties, a person reaches skeletal maturity [7]. By then, all the cartilage has been replaced by bone, and no further growth in bone length is possible. This mechanism is highly regulated and requires continuous signal interchange between cartilage and bone cells.

The GP is a functionally and cytologically specialized region consisting of three tissue types: the cartilage, which is divided into four distinct zones, the bony tissue of the metaphysis, and the vascular network surrounding the GP. The close interaction between these components is critical for normal bone formation and growth (Figure 3).

In the GP, chondrocytes are organized into vertical columns (Figure 2) and undergo a carefully regulated differentiation process critical for endochondral ossification, which drives longitudinal bone growth. This process of differentiation occurs in synchronized stages, with the chondrocytes at each horizontal level in different columns maturing simultaneously, resulting in a layered structure within the GP. Structurally and functionally, the growth plate is divided into four zones, each characterized by chondrocytes at distinct stages of differentiation: resting (RCs), proliferative (PCs), prehypertrophic (PHCs), and hypertrophic (HCs). In the resting zone, the chondrocytes are small, spherical, and relatively inactive, serving as a reservoir for future proliferating cells. Chondrocytes in the proliferative zone then become flattened, align into columns, and undergo active mitosis. As chondrocytes progress to the PHC and HC zones, they exit the cell cycle, increase in size, and undergo hypertrophy. In this stage, HCs contribute to extracellular matrix changes that prepare the matrix for mineralization. They also play a key role in signaling for vascular invasion, which brings blood vessels into the calcified cartilage matrix to supply nutrients and introduce osteoprogenitor cells to the ossification front (Figure 2B and Figure 3). This vascularization process enables osteoblasts to replace the calcified cartilage with bone tissue (Figure 2B,C). Simultaneously, osteoclasts resorb portions of the calcified cartilage, clearing space for bone deposition and remodeling the matrix to sustain continuous growth. Through this coordinated effort between chondrocytes, osteoblasts, osteoclasts, and endothelial cells, the growth plate promotes bone elongation, ensuring proper skeletal development.

### 2.2. Structure of the Bone

Bone tissue is a dynamic, highly organized structure that not only provides support, locomotion, and protection for vital organs but also functions as an endocrine organ. Bones are also designed to bear weight and distribute forces across the body. Their structure varies depending on the bone type and function, from dense, compact bone in areas needing strength (Figure 4) (like the diaphysis of long bones) to spongy bone in areas requiring some flexibility and shock absorption (like in the epiphysis). They play a central role in regulating mineral balance, acid–base homeostasis, and blood cell production (hematopoiesis). Comprised of distinct layers and components, each with specialized roles, bone tissue is essential for growth, remodeling, and repair (Figure 4). Although rigid, bone tissue is in a continuous state of renewal, with the entire human skeleton being replaced approximately every 7–10 years [8]. Examining the interaction of osteoblasts, osteocytes, and osteoclasts within this environment sheds light on the complexity of bone tissue formation, structuring, and renewal. Therefore, bones function as structural and load-bearing organs that adapt to stress and facilitate movement. Growth plates enable bones to grow in length, distribute loads, and adapt to controlled mechanical stress, all of which are essential for proper skeletal development in growing individuals. Proper function and interaction between bones and growth plates are crucial for a balanced and healthy musculoskeletal system. To fully appreciate these functions, it is essential to examine the unique structural organization that underpins their remarkable strength, adaptability, and role in growth.

The outermost covering of bone, the periosteum, consists of two layers: a fibrous outer layer and an inner cellular, vascular layer capable of bone formation. Beneath this lies the cortical or compact bone, a dense layer that forms the external wall of bone cavities. Cortical bone is especially prominent in the diaphysis (shaft) of long bones, being supported by a dual blood supply that sustains the tissue even if one source is compromised. Cortical bone is organized into osteons, which are cylindrical structures housing osteocytes within a matrix rich in calcium and phosphorus, primarily in the form of hydroxyapatite. Collagen fibers within this matrix provide the bone with a degree of flexibility, enhancing its durability and resilience.

Trabecular or cancellous bone is found primarily at the ends of long bones (the metaphysis and epiphysis) and has a porous, lattice-like structure that maximizes its strength while minimizing its mass. This three-dimensional architecture is essential for distributing loads and absorbing stress efficiently. In the epiphysis, or the enlarged ends of long bones, trabecular bone provides additional support and flexibility. The disruption of this trabecular organization can lead to deformities, a reduced ability to adapt quickly to mechanical demands, and a greater susceptibility to fractures [9].

The organization of bone tissue into mineralized and non-mineralized regions, along with a network of spaces that facilitate cellular communication, enables each cell type to perform its role in bone remodeling. By understanding bone structure, we gain valuable insights into how bone cells coordinate to maintain bone integrity, repair damage, and adapt to shifting mechanical requirements.

### 2.3. Cellular Components and Their Roles in Bone Growth

Bone growth and remodeling involve the coordinated activities of multiple cell types, including chondrocytes, osteoblasts, osteoclasts, osteocytes, and vascular cells. These cells engage in complex signaling interactions that ensure proper bone formation and maintenance (Figure 5 and Table 1).

Chondrocytes: As the primary cells within the growth plate, chondrocytes are responsible for producing and maintaining the cartilage matrix. During endochondral ossification, they progress from a proliferative phase to hypertrophy and eventually apoptosis. Key markers include *SOX9*, *COL2A1*, and *ACAN* in early development and *COL10A1, RUNX2*, and *MMP13* during hypertrophy and bone formation. The balance of chondrocyte proliferation and differentiation is finely regulated by signaling molecules such as *Ihh* (Indian hedgehog) and PTHrP (parathyroid hormone-related protein), allowing chondrocytes to communicate with other cells in the bone microenvironment [10].

Osteoblasts: Osteoblasts are bone-forming cells derived from plenipotentiary multipotent stromal cells (MSCs) [11]. They are cuboidal cells that are located along the bone surface, comprising 4–6% of its total, and are largely known for synthesizing the organic matrix of bone, including collagen, and facilitating its subsequent mineralization [12]. Their differentiation follows a progressive sequence starting from skeletal stem cells and moving through pre-osteoblast stages, being marked by the expression of specific transcription factors and proteins. In the early stages of differentiation, osteoblasts express markers such as *RUNX2* and osterix (Osx), which are essential for the commitment to the osteoblast lineage. As they transition to matrix-producing osteoblasts, the cells begin expressing *collagen type I* (*Col1*), alkaline phosphatase (ALPL), and osteocalcin (OCN or previously called Bglap), which are key proteins involved in matrix production and early mineralization. In the mature stage, OCN continues to be expressed, alongside *PHOSPHO-1* and the parathyroid hormone receptor 1 (*PTH1R*), which contribute to the regulation of bone remodeling. During mineralization, osteoblasts express additional markers such as matrix extracellular phosphoglycoprotein (*MEPE*) and dentin matrix protein 1 *(DMP1*), phosphate-regulating endopeptidase homolog X-linked (*PHEX*), sclerostin (*SOST*), and fibroblast growth factor 23 (FGF23), which modulate the deposition of minerals into the bone matrix. These markers not only reflect the osteoblasts’ functional stages but also their role in maintaining the balance between bone formation and mineralization [13].

Osteoclasts: Osteoclasts are multinucleated cells derived from hematopoietic stem cells that are responsible for bone resorption. As polarized cells, they are responsible for breaking down bone and cartilage tissue, a function that is essential for the growth, maintenance, repair, and remodeling of bones. They degrade the mineralized bone matrix during bone remodeling, which is essential for maintaining bone homeostasis. Markers and key molecules include RANK (receptor activator of nuclear factor kappa-B), TRAF6 (TNF receptor-associated factor), cathepsin K (*CTSK*), and tartrate-resistant acid phosphatase (TRAP). The activity of osteoclasts is particularly important in the remodeling phase of endochondral ossification, where they resorb the calcified cartilage matrix, allowing for the deposition of new bone [14].

Osteocytes: Osteocytes are mature osteoblasts that become embedded within the bone matrix. They are the most numerous cells in the bone, representing 90–95% of the total amount, as well as the most long lived, with a lifespan of up to 25 years. Osteocytes are spider-shaped cells that are buried in the mineralized bone matrix, originating from the mesenchymal stem cell (MSC) lineage through osteoblast differentiation [15]. Each osteocyte has up to 50 long and 450 branched cellular processes that extend through a network of interconnected canaliculi within the bone. These cells act as mechanosensors, detecting changes in mechanical strain and signaling osteoblasts and osteoclasts to adjust bone formation and resorption. Osteocytes classical markers are *osteopontin (OPN*), *DMP1*, *bone sialoprotein (BSP*) and *OPG*. They also express factors that inhibit bone formation, such as sclerostin, *FGF23*, and *MEPE* [14].

Osteoprogenitors: These stem cells, located in the bone, play a pivotal role in bone repair and growth. Derived from MSCs, they are precursors to the more specialized bone cells such as osteocytes and osteoblasts. They reside in the bone marrow, and their numbers decrease with age [16].

Vascular cells: The invasion of blood vessels into the hypertrophic zone of the growth plate is a critical step in endochondral ossification. Vascular cells, including endothelial cells, are attracted by molecules secreted by HCs. This vascular invasion delivers nutrients, osteoprogenitor cells, and other factors necessary for ossification. The vascularization of human growth plates was studied via immunohistochemistry using specific endothelial cells markers such as CD34 and CD31. Markers for endothelial progenitor cells can be CD133, CXCR4, and vascular endothelial growth factor receptor 2 (VEGFR-2). The interplay between vascularization and bone formation is essential for proper skeletal development.

Finally, a comprehensive understanding of cell types and their specific markers is essential for elucidating the cellular interactions that regulate bone and cartilage growth and repair. Distinct cell types, including chondrocytes, osteoblasts, and osteocytes, play specialized roles in tissue development and regeneration. Identifying unique markers for each cell type enables the precise tracking of their functional states and provides insights into the coordinated mechanisms driving growth and repair. This knowledge is fundamental for designing targeted therapeutic strategies aimed at enhancing cartilage health and improving tissue regeneration outcomes.

## 3. Signaling Pathways and Cell–Cell Interactions in Bone Growth

### 3.1. Local Signaling Pathways Controlling Bone Growth

Local signaling pathways are responsible for regulating the process of bone growth by integrating the activities of the cells within the bone microenvironment (Figure 5). These pathways include numerous molecular signals that mediate communication between chondrocytes, osteoblasts, osteoclasts, and osteocytes, orchestrating the complex processes of cell differentiation, matrix production, and tissue remodeling. Together, these local pathways ensure that the growth and remodeling of bone are tightly regulated to meet the changing physiological needs throughout development and into adulthood.

#### 3.1.1. *Ihh*-PTHrh Feedback Loop in Growth Plate Development

The *Ihh*-PTHrP feedback loop has been extensively reported to play a critical role in regulating chondrocyte proliferation and differentiation. Embryonic proliferative chondrocytes in the GP express PTHrP and contribute to the formation of both the articular cartilage and GP structures. Around the sixth and seventh weeks of embryonic development, the secondary ossification center (SOC) appears. At this stage, PTHrP is detected in the upper zone of the GP (resting zone) and in the superficial region of the AC. PTHrP and its receptor are believed to delay chondrocyte hypertrophy, preventing premature mineralization. Conversely, *Ihh* produced by PHC has an opposite effect: it inhibits proliferation while promoting final differentiation and hypertrophy [17]. This interaction forms the PTHrP-*Ihh* feedback loop, which is critical for maintaining the structure of the GP. Disruption of either the PTHrP gene or the PTH/PTHrP receptor gene accelerates the differentiation of the GP chondrocytes, leading to increased osteoblast numbers, excessive matrix accumulation, and reduced vascular invasion [18,19]. Moreover, *Ihh* controls the differentiation of osteoblasts and designates the location at which this differentiation occurs [20,21].

#### 3.1.2. Wnt Signaling

The Wnt signaling pathway, encompassing the canonical WNT/β-catenin pathway as well as the non-canonical WNT/PCP and WNT/Ca^2^⁺ pathways, plays a pivotal role in regulating chondrocyte and osteoblast differentiation.

In the canonical pathway, WNT ligand interactions with LRP5/6 and frizzled receptors result in the stabilization of β-catenin. The stabilized β-catenin translocates to the nucleus to activate genes that determine cell proliferation and differentiation in chondrocytes and osteoblasts. When WNT signaling is absent, β-catenin is degraded, leading to the suppression of these processes. The canonical pathway is particularly crucial for the transition of mesenchymal progenitors into mature osteoblasts, with WNT activation correlating with increased bone mass. Therefore, proper regulation is necessary for normal chondrocyte hypertrophy, while disruptions in these pathways affect both chondrocyte proliferation and osteoblast differentiation, highlighting the essential role of WNT signaling in skeletal development [22].

The non-canonical WNT/PCP pathway activates JUN kinase (JNK), a process essential for chondrocyte column formation in the growth plate, while the WNT/Ca^2^⁺ pathway triggers calcium release and nuclear factor of activated T-cells (NFAT) nuclear translocation to drive gene expression. Without JNK activation, chondrocytes fail to align properly, leading to disorganized growth plates. This misalignment can result in abnormal bone elongation and potentially shorter bones, as the structure of the GP is essential for directional growth. The WNT/Ca^2^⁺ pathway also has significant effects on osteoblasts, where it activates genes involved in cell differentiation and function [23]. WNT signaling has a dual role in bone development, inhibiting chondrogenesis through the repression of *SOX9*, a key transcription factor for all phases of the chondrocyte lineage, while promoting chondrocyte hypertrophy and osteoblast differentiation. Since *SOX9* is essential for chondrocyte lineage progression, excessive WNT activity can result in a failure to form or maintain sufficient cartilage tissue, impairing early bone formation and potentially leading to conditions like shortened bones or reduced cartilage in joints [24,25].

In the absence of proper Wnt signaling, growth may result in stunted or skeletal abnormalities due to the lack of cell proliferation, differentiation, and chondrocyte alignment needed for good bone and cartilage development. In this context, crosstalk between Wnt and PTHrP signaling ensures a coordinated transition from cartilage to bone. Wnt signaling can enhance the expression of *Ihh*, which stimulates the PTH signaling pathway, playing a crucial role in regulating chondrocyte activity. Similarly, PTH-*Ihh* signaling can modulate Wnt target genes, with *Ihh* promoting the expression of Wnt ligands, establishing a feedback loop that reinforces chondrocyte maturation [26]. Thus, both pathways contribute significantly to the transition from cartilage to bone during endochondral ossification. Ultimately, Wnt signaling enhances osteoblast function, while PTH-*Ihh* signaling ensures the proper progression of chondrocyte maturation necessary for growth plate development. *RUNX2*, a transcription factor essential for the transition of chondrocytes from the proliferative stage to the hypertrophic stage, appears to influence Wnt and *Ihh* signaling together, upregulating downstream factors to enhance chondrocyte proliferation by activating both pathways [27].

#### 3.1.3. TGF-β/BMP Signaling

Transforming growth factor-beta (TGF-β) and *bone morphogenetic proteins (BMPs)* are pivotal in regulating various processes in the growth plate during endochondral ossification. Both TGF-β and *BMPs* function through both canonical (Smad-dependent) and non-canonical (Smad-independent) pathways. These signaling pathways are essential for controlling osteoblast differentiation from mesenchymal stem cells. Beyond Smad signaling, non-canonical pathways such as *p38 MAPK* also intersect at *RUNX2*, which also acts as a driver for osteoblast differentiation [28].

During chondrogenesis, TGF-β and BMP signaling are crucial for the expression and maintenance of *SOX9*, which is necessary for chondrogenic mesenchymal condensations. TGF-β promotes mesenchymal cell condensation and chondrocyte proliferation but inhibits the terminal differentiation of chondrocytes. In contrast, BMP signaling, particularly BMP-2, BMP-4, and BMP-7, is indispensable for mesenchymal differentiation, as well as chondrocyte proliferation and maturation. BMP inhibitors accumulate in the resting zone to keep cells in a relatively static state [29]. TGF-β and BMPs also stimulate the production of extracellular matrix components, such as collagen types II (Col II) and X and *ACAN* [30]. The unique expressions of BMPs establish a signaling gradient in the growth plate that promotes chondrocyte maturation (Figure 4). Moreover, *Ihh* enhances BMP expression, while BMP signaling also regulates *Ihh* expression, resulting in a feedback loop [31].

BMP signaling, particularly through BMP-2, BMP-4, BMP-5, BMP-6, and BMP-7, plays a pivotal role in osteogenesis. BMP-2 and BMP-4 coordinate the transition from *RUNX2*-positive to OSX-positive osteoblasts, while BMP-7 enhances osteoblast differentiation and calcium mineralization. However, BMP-3 acts as a negative regulator, inhibiting the differentiation of skeletal progenitor cells into mature osteoblasts and influencing adult bone mass [22,32].

Chen et al. showed that elevated Wnt3a levels enhance BMP2 and BMP4 expression in chondrocytes, suggesting that Wnt/β-catenin signaling may activate BMP signaling. thereby promoting chondrocyte differentiation by stimulating BMP pathways [33]. Additionally, Wnt/β-catenin signaling regulates BMP7, as inhibiting this pathway leads to reduced BMP7 expression, which, in turn, diminishes the differentiation of mesenchymal stem cells into chondrocytes [34].

Furthermore, TGF-β/*BMP* signaling engages in intricate crosstalk with other critical pathways, including hedgehog, notch, and fibroblast growth factors (FGFs). These interactions fine-tune the regulation of osteoblast differentiation at various stages, highlighting the complexity and coordination required for proper bone development and maintenance [28].

#### 3.1.4. FGF Signaling

FGFs generally antagonize the effects of BMPs during chondrogenesis. Relevant genetic studies have shown that FGF signaling crucially regulates chondrocyte proliferation and differentiation. Many of the 22 distinct FGF genes and four FGF receptor genes are expressed at every stage of endochondral bone formation [10]. FGF18 helps regulate chondrocyte proliferation and differentiation, influencing the timing of hypertrophy and ossification [35]. *FGFR3* in chondrocytes restricts their proliferation and accelerates their hypertrophic differentiation [36].

FGF signaling also has a significant impact on osteoblast differentiation. FGF-2 knockout mice showed a decreased bone mass coupled to an increase in adipocytes in the bone marrow, indicating the participation of FGFs in the osteoblast differentiation [37]. It has also been demonstrated that FGF-18 upregulates osteoblast differentiation via an autocrine mechanism [15].

There are other FGFs, such as FGF23 and FGF2, that are also known as basic FGFs (bFGFs). FGF23 plays a crucial role in phosphorus and calcium metabolism, indirectly affecting bone health [38,39]. FGF18 and FGF2 are both secreted by chondrocytes; however, FGF23 is primarily secreted by osteoblasts and osteocytes. FGF23 influences the overall mineral environment, including phosphate levels, which can affect chondrocyte function. FGF23 can affect chondrocytes in the growth plate, where it may modulate the balance between proliferation and hypertrophy [40]. This is critical for the proper development of bone and its final length. Other members of the FGF family, such as FGF1, also contribute to proliferation and regeneration processes in bone and cartilage tissues. The interaction and balance between these factors are fundamental for proper bone development and remodeling [41].

#### 3.1.5. IGF1 Signaling

Insulin-like growth factor I (IGF-I) is crucial for bone growth, functioning through both local paracrine effects within the growth plate and systemic endocrine pathways. Locally, chondrocytes, especially those in the proliferative and hypertrophic zones, produce IGF-I to stimulate their own proliferation and maturation, driving longitudinal bone growth. Osteoblasts are the primary source of IGF-I in bone. They produce IGF-I, which acts in an autocrine and paracrine manner to promote their own differentiation and activity, through the regulation of Ephrin B2/EphB4 pathways, as well as to regulate osteoclastogenesis. Additionally, IGF-I regulates the balance between bone formation and resorption by modulating the production of *RANKL* and OPG, which controls osteoclast differentiation and activity. In osteoblasts, IGF-I enhances matrix deposition and mineralization, contributing to bone density and structural integrity. This local signaling is essential for coordinating skeletal development and maintaining bone health [42].

On a systemic level, growth hormone (GH) is a primary regulator of IGF-I production. GH is secreted by the anterior pituitary and stimulates IGF-I synthesis primarily in the liver, which then releases IGF-I into the bloodstream. This circulating IGF-I contributes to growth by promoting the proliferation of chondrocytes and osteoblasts in bones and affects growth across multiple organ systems [43].

### 3.2. Cell–Cell Interactions

The investigation of growth plate signaling pathways is undeniably crucial for understanding bone development. However, the study of cell–cell interactions, particularly between chondrocytes and cells of the osteogenic lineage, presents an even more compelling and less-explored dimension of this process (Figure 6). These interactions are pivotal for coordinating the complex events of endochondral ossification. Recent findings have highlighted that HCs not only regulate the differentiation of osteoblasts and osteoclasts but also engage in bidirectional communication with these cells. Understanding how these cellular interactions influence each other could reveal new insights into the mechanisms underlying both normal growth conditions and pathogenic situations in the search for new therapeutic targets.

#### 3.2.1. Chondrocytes–Endothelial Cells

Vascular cells play a critical role in supporting growth plate development by delivering essential nutrients and signaling molecules to the hypertrophic zone, which influences chondrocyte hypertrophy [29]. Chondrocytes in the growth plate operate under hypoxic conditions, especially because cartilage is inherently avascular. The transcription factors hypoxia-inducible factors (HIFs) (especially HIF-1α) are activated in response to hypoxia, leading to the expression of genes that adapt chondrocytes to low oxygen, including those involved in anaerobic metabolism. One of the key target genes of HIF-1α is *vascular endothelial growth factor* (*VEGF*), produced by HCs in the growth plate [44] (Figure 4). VEGF binds to VEGFR-1 and VEGFR-2 receptors on endothelial cells, stimulating their proliferation, migration, and tube formation, thereby promoting vascularization [45]. There are different VEGF isoforms (VEGF120, VEGF164, and VEGF188) secreted by chondrocytes, each with varying diffusion capacities. For example, VEGF120 is highly diffusible, while VEGF188 binds tightly to the extracellular matrix [46]. The isoforms help guide endothelial cells toward the hypertrophic cartilage, ensuring that vascularization occurs at the right time and place. Mouse models deficient in either HIF-1α or VEGF show severe defects in vascularization, leading to impaired endochondral ossification and dwarfism [36]. Vascularization also facilitates the recruitment of osteoblast and osteoclast precursor cells to the hypertrophic zone, as well as the induction of apoptosis in HCs [47,48]. In mice deficient in HIF-1α, a protein that increases significantly under low oxygen conditions [49,50], there is a marked increase in chondrocyte apoptosis in the central epiphyseal regions of developing cartilage [50]. The overexpression of VEGF partially rescues this phenotype by reducing chondrocyte apoptosis, underscoring VEGF’s crucial role in promoting chondrocyte survival [51]. This mechanism is important because cartilage is avascular, and HCs require increased nutrients and oxygen. Vascular invasion into the hypertrophic zone of the growth plate meets these demands, supporting chondrocyte survival during early hypertrophy and facilitating apoptosis when they are ready for replacement by bone cells. This process is essential for the removal of cartilage and the establishment of ossified bone. Proper regulation of these processes is critical for normal bone growth and development. Nevertheless, VEGF signaling shares pathways with growth factors like EGF and PDGF, leading to overlapping gene regulation, particularly through receptor tyrosine kinases (RAS-Raf-ERK1/2 and PI3K-Akt), which makes its study challenging. Even so, the gene profiles regulated by VEGF in other cell types, such as mesenchymal progenitors, osteoblasts, or chondrocytes, remain poorly understood. In addition to VEGF, chondrocyte-derived factors such as matrix metalloproteinases (MMPs), particularly *MMP13*, contribute to the breakdown of the cartilage matrix, clearing the way for vascular invasion [52]. This process is necessary for the removal of hypertrophic cartilage and allows for the migration of vascular endothelial cells and osteoclasts. VEGF signaling also promotes the survival and migration of endothelial cells by interacting with its receptor, VEGFR2, which is expressed on the surface of vascular cells [53].

Therefore, endothelial cells invading the hypertrophic cartilage do more than deliver oxygen and nutrients. They also release angiocrine signals that promote osteogenesis and may even regulate chondrocyte fate. For example, endothelial cells can influence the expression of noggin, which modulates BMP signaling, thereby affecting the coupling of angiogenesis and osteogenesis.

#### 3.2.2. Chondrocytes–Osteoblasts and Cell Transdifferentiation

The differentiation of chondrocytes and osteoblasts is tightly linked. As chondrocytes progress through different stages of maturation in the GP, they signal to the surrounding perichondrium to trigger osteoblast differentiation. This process occurs in several phases including osteoblast proliferation, extracellular matrix maturation, and subsequent matrix mineralization [54]. Chondrocytes, particularly HCs of the growth plate, release specific signaling molecules that promote osteoblast differentiation and activity. One such molecule is type X collagen, which is produced to establish a calcified matrix that serves as a scaffold for osteoblasts to initiate bone deposition. In addition, the BMP molecules released by HCs further support osteoblast differentiation and activity by enhancing their maturation and matrix production. *Ihh*, another pivotal signaling molecule in this crosstalk, directly stimulates the differentiation of perichondral progenitors into osteoblasts. *Ihh* stimulates *RUNX2* expression in mesenchymal progenitor cells in the perichondrium, driving their differentiation into osteoblasts. *Ihh* signaling thus coordinates the timing of bone collar formation (the initial mineralized bone structure) with the maturation of chondrocytes in the growth plate.

Osteoblast differentiation is also driven by IGF-I through the activation also of *RUNX2* and OSX [55,56]. OSX acts downstream of *RUNX2* to further drive the differentiation of osteoprogenitors into functional osteoblasts. This signaling cascade ensures that osteoblast differentiation is synchronized with the hypertrophic differentiation of chondrocytes [57]. Recent in vitro functional studies showed that Sox6, which is expressed by proliferative chondrocytes in the GP, promotes the multiplication of osteoblasts [58]. Moreover, hypertrophic cells were also found to secrete a C-type lectin domain protein, Clec11a, which promotes osteogenesis [59], further illustrating the bidirectional crosstalk between chondrocytes and osteogenic cells [59].

Osteoblast lineage cells in the perichondrium can also signal back to chondrocytes to regulate their differentiation and activity. These osteocrine signals include secreted molecules like Wnt, BMPs, and FGFs, which can affect chondrocyte behavior. For example, the BMPs secreted by osteoblasts stimulate chondrocyte proliferation and inhibit their hypertrophic differentiation, while FGFs, particularly FGF18, acts to accelerate the maturation of chondrocytes into HC [36].

A novel player in chondrocyte–osteoblast communication is pannexin 3 (Panx3), which facilitates cell-to-cell communication by functioning as a gap junction protein and hemichannel. Panx3 allows for the transfer of ions and small signaling molecules like ATP and Ca^2^⁺, which are crucial for their coordinated differentiation. In chondrocytes, Panx3 promotes their maturation into HCs and regulates vascular invasion. In osteoblasts, Panx3 modulates intracellular Ca^2^⁺ levels and activates signaling pathways, such as Wnt/β-catenin, which are critical for the bone deposition and mineralization processes [60].

WNT proteins, which are secreted by both chondrocytes and osteoblasts, play a pivotal role in coordinating their communication. Canonical Wnt signaling in osteoblasts promotes their differentiation and enhances bone formation. Meanwhile, chondrocytes produce Wnt signals that regulate their own differentiation while stimulating the proliferation of neighboring osteoblasts [61]. Additionally, FGF18, which is secreted by osteoblasts, acts on chondrocytes to negatively regulate their proliferation in the GP by activating *FGFR3*, thereby controlling the transition from proliferation to hypertrophy.

One of the most fascinating aspects of chondrocyte–osteoblast interactions is the recent discovery that chondrocytes can transdifferentiate into osteoblasts. Contrary to earlier beliefs that HCs ultimately undergo apoptosis, evidence suggests that some HCs re-enter the cell cycle and activate osteogenic genes, allowing them to contribute directly to new bone formation [62,63]. This process occurs in the transition zone, located adjacent to the hypertrophic zone, where HCs switch off their chondrogenic program and begin expressing osteogenic markers such as *RUNX2* and Osx [63,64]. Genetic lineage tracing studies using Col10a1-Cre and Agc1-CreERT2 have confirmed that labeled chondrocytes can function as osteoblasts, integrating into trabecular bone surfaces and synthesizing new bone matrix [65]. Yang et al. [56] genetically labeled either HCs by Col10a1-Cre or chondrocytes by tamoxifen-induced Agc1-CreERT2 using enhanced green fluorescent protein (EGFP), LacZ, or Tomato expression. Both Cre drivers were specifically active in chondrocytic cells and not in the perichondrium, in the periosteum, or in any of the osteoblast lineage cells. After labeling, these cells were distributed throughout trabecular surfaces and, in the endosteum, embedded within the bone matrix. In vitro studies demonstrated that a proportion of the non-chondrocytic cells derived from chondrocytes labeled by *COL10A1*-Cre or Agc1-CreERT2 were functional osteoblasts. Bmpr1a (BMP receptor 1a) has been described to be critical for maintaining the pool of chondrocytes that can transform into bone-forming cells. The knockout of Bmpr1a in chondrocytes results in severe defects in skeletal development, including malformed epiphyses, absence of metaphyses, and reduced cortical bone thickness, which highlights that chondrocyte-derived cells continue to contribute to bone remodeling in adulthood [66].

Thus, the crosstalk between chondrocytes and osteoblasts is not limited to signaling molecules but also includes direct cell lineage transitions, providing a more dynamic and integrated model of skeletal development and maintenance.

#### 3.2.3. Chondrocytes–Osteoclasts

Osteoclasts break down the newly formed bone to open up the medullary cavity and trabecular formation promotes bone length. As we presented before, several studies pointed out that osteocytes produce cytokines as receptor activators of *RANKL*, essential for osteoclast differentiation, function, and survival in mature bone [67,68,69] (Figure 1). Recent studies have demonstrated that HCs, like osteoblasts, also express both *RANKL* and OPG, suggesting that the GP actively participates in the regulation of bone resorption. In addition, matrix metalloproteinases, particularly MMP-13, secreted by HCs, facilitate osteoclast activity by preparing the cartilaginous matrix for degradation. This preparation enables osteoclasts to efficiently resorb the cartilage and bone matrix [70]. IGF-I, expressed in chondrocytes, also enhances *RANKL* expression, further promoting osteoclastogenesis and linking IGF-I signaling to bone resorption processes within the growth plate [42].

Moreover, the epidermal growth factor receptor (EGFR) pathway has recently been implicated in cartilage–bone cell communication. EGFR activation in chondrocytes induces the production of MMPs and *RANKL* (Figure 4). These molecules interact with osteoblasts and other bone cells, playing key roles in bone remodeling and cartilage–bone crosstalk. Additionally, α-parvin, an integrin-associated focal adhesion protein, is crucial for regulating the orientation of chondrocytes during columnar formation in the growth plate [71]. In animal models, the loss of α-parvin leads to increased binucleation, higher rates of cell death, and the dilation of the resting zones in mature growth plates [72].

#### 3.2.4. Chondrocytes–Osteocytes

The interaction between chondrocytes and osteocytes is critical in maintaining bone homeostasis and facilitating growth. Chondrocytes communicate with osteocytes through molecular signaling, which influences both bone and cartilage remodeling processes.

Sclerostin, being primarily secreted by osteocytes, plays a pivotal role in regulating the interaction between osteocytes and chondrocytes, particularly in the context of bone formation and remodeling. By inhibiting the Wnt/β-catenin pathway, sclerostin suppresses osteoblast activity, thus negatively regulating bone formation. Interestingly, while sclerostin is mainly secreted by osteocytes, it has also been found in chondrocytes [73] and osteoclasts [74] at certain stages of skeletal development, including during chondrocyte hypertrophy.

One of the key factors involved in this crosstalk is IGF-1, which is transported from osteocytes into the cartilage through diffusion across a concentration gradient, a process that is enhanced by mechanical loading. This interaction promotes growth and chondrocyte proliferation in healthy bones [75]. Moreover, osteocytes also secrete FGF-23, which has been shown to inhibit chondrocyte proliferation and maturation via the *Ihh* pathway, further demonstrating the regulatory role of osteocytes in cartilage development [76].

#### 3.2.5. Osteoblasts–Osteoclasts

To understand how growth is orchestrated at the cellular level, it is crucial to consider the interactions not only between chondrocytes and other cells but also between bone cells. The communication between osteoblasts and osteoclasts plays a significant role in regulating bone turnover and provides essential feedback that affects chondrocyte activity and cartilage remodeling. For example, osteoblasts produce key molecules, such as *RANKL* and M-CSF, that influence osteoclast differentiation while also secreting OPG to regulate this process. OPG, a decoy receptor that binds *RANKL*, prevents its interaction with RANK and thereby inhibits osteoclast maturation [77]. This finely tuned regulation maintains the balance of bone resorption and formation, indirectly influencing the growth plate and chondrocyte-mediated growth [15,16].

In this system, both membrane-bound *RANKL* (m*RANKL*) and soluble *RANKL* (s*RANKL*) play distinct roles. m*RANKL* requires direct cell-to-cell contact to activate osteoclastogenesis, whereas s*RANKL* can act over a distance, diffusing to bind RANK on osteoclast precursors independently of direct contact. Reverse signaling in osteoblasts, triggered by vesicular RANK from mature osteoclasts, further activates pathways like PI3K-mTORC1 to promote osteoblast differentiation and bone formation [78].

Furthermore, factors such as IGF-I play a dual role in promoting osteoclastogenesis and regulating bone remodeling, as well as influencing chondrocyte proliferation and differentiation in the growth plate. IGF-I is known to stimulate *RANKL* production in osteoblasts, promoting osteoclast activity for bone resorption. However, IGF-I also enhances cartilage matrix production by chondrocytes, linking cartilage growth to the remodeling of adjacent bone [42].

Other pathways, such as EphrinB2-EphB4 and Wnt5a-Ror2, mediate bidirectional signaling that coordinates bone and cartilage cells’ activities to ensure that growth plate cartilage is progressively replaced by bone. EphrinB2 produced by osteoclast binds to EphB4 on osteoblasts, inhibiting osteoclast formation while stimulating osteoblast differentiation [79]. Additionally, osteoclast-derived Semaphorin 4D (Sema4D) can inhibit osteoblast activity, which indirectly influences the environment of chondrocytes by modulating bone remodeling near the growth plate [80]. In contrast, osteoclast-derived Semaphorin 4D (Sema4D) binds to Plexin-B1 on osteoblasts, inhibiting bone formation by suppressing IGF-1 signaling, providing a feedback mechanism from osteoclasts to regulate osteoblast activity [81]. Together, these signaling interactions between osteoblasts, osteoclasts, and chondrocytes create a complex communication network essential for endochondral ossification and bone growth. By regulating both cartilage expansion and bone remodeling, these pathways ensure that the growth plate functions effectively, converting cartilage into bone to promote longitudinal growth. In cases where these pathways are disrupted, chondrocyte function may be impaired, leading to growth disorders like short stature.

#### 3.2.6. Osteoblasts–Osteocytes

Osteocyte–osteoblast interactions influence the bone environment around cartilage, which, in turn, affects chondrocyte behavior. The osteocyte–osteoblast communication network plays a pivotal role in adjusting bone formation in response to mechanical and biochemical cues. Osteocytes and osteoblasts maintain a dynamic communication network that regulates bone formation and remodeling. Osteocytes, the primary mechanosensors embedded in the bone matrix, produce sclerostin, which inhibits bone formation by binding to LRP4/5/6 and antagonizing the Wnt/β-catenin signaling pathway, thereby suppressing early osteoblast differentiation. This inhibitory function is essential for preventing excessive bone deposition, especially under a low mechanical load, ensuring that bone growth does not exceed the structural needs of the surrounding tissues. In contrast, under mechanical strain, osteocytes secrete IGF-I, which acts in a paracrine manner on osteoblasts, promoting their differentiation and bone-forming activity [22]. By producing IGF-I in response to load, osteocytes indirectly support chondrocyte activity in the growth plate by ensuring a balanced and adaptive bone environment conducive to endochondral ossification [75].

Furthermore, the osteocyte network utilizes gap junctions to facilitate intercellular communication, especially via connexin 43 (Cx43). This gap junction protein enables the direct transfer of signaling molecules and ions between osteocytes and osteoblasts, coordinating their activities to balance bone resorption and formation. Mechanistically, Cx43 hemichannels also modulate osteocyte responses to microenvironmental changes, enhancing the secretion of bone-promoting factors like prostaglandins and ATP under mechanical strain [82]. Mechanosensitive Cx43 hemichannels in osteocytes also respond to changes in the mechanical load by releasing prostaglandins and ATP, which can have downstream effects on the bone–cartilage interface, supporting the structural adaptation needed for continuous bone growth.

While osteocyte–osteoblast signaling is essential for endochondral ossification and bone remodeling, its influence extends beyond growth processes and impacts various cartilage-related disorders, including osteoarthritis (OA) [79]. In addition to supporting growth plate function, the bone microenvironment created by osteocyte–osteoblast interactions plays a significant role in maintaining cartilage integrity at the joints. In OA, excessive mechanical stress or inflammation can disturb osteocyte function, increasing the production of sclerostin and inflammatory cytokines. This elevated sclerostin level inhibits the Wnt/β-catenin pathway, suppressing osteoblast differentiation and promoting bone resorption over formation [83]. Disruptions in these pathways can contribute to the degeneration of cartilage, as seen in OA [84], where the interplay between bone and cartilage cells becomes imbalanced, leading to structural and biochemical changes that degrade joint health. Additionally, changes in Cx43 signaling have been linked to increased OA and subchondral bone sclerosis [85]. These Cx43-mediated changes impact chondrocytes by modifying the load distribution across the joint, contributing to cartilage breakdown and the progression of osteoarthritic changes.

While osteocyte–osteoblast communication is vital for maintaining a supportive environment for chondrocyte-mediated growth, the dysregulation of these pathways can contribute to cartilage-related disorders like osteoarthritis. This intricate signaling network thus serves not only as a foundation for normal bone and cartilage function but also as a potential therapeutic target for conditions involving both growth impairments and degenerative joint disease.

## 4. Disorders of Bone Growth: Clinical Implications and Future Perspectives

Many human skeletal disorders are caused by abnormalities in the endocrine and paracrine regulatory systems and the cellular interactions that control bone growth and remodeling. These disorders can arise due to genetic mutations, injuries, chronic diseases, or metabolic conditions that affect the GP and bone development [54,55]. Due to the extensive spectrum of these conditions, only a subset can be discussed in detail herein. Additional disorders are summarized in Table 2 to provide a more comprehensive perspective on the range of affected signaling pathways and their clinical manifestations.

Disruptions in PTH/PTHrP signaling can arise from genetic mutations, altered receptor function, or dysregulated expression. These disruptions affect the cartilage–bone crosstalk in several ways. PTHrP normally prevents chondrocytes from undergoing hypertrophy too early. Without functional PTHrP signaling, chondrocytes become hypertrophic prematurely, accelerating the process of endochondral ossification. This leads to early ossification of growth plates, stunting bone growth and leading to skeletal abnormalities, such as shorter bones and altered bone shape. Mutations leading to defective or absent PTH/PTHrP receptors have been found in human fetuses with Blomstrand chondro-osteodystrophy, a condition characterized by prenatal death and shortened limbs with premature ossification [87]. Another example can be Jansen Metaphyseal Chondrodysplasia (JMC) [53], an autosomal dominant disease characterized by mutations that lead to the constitutive activation of the PTH/PTHrP receptor (*PTH1R*) in the GP (Figure 1), resulting in an aberrant cartilage-to-bone transition that disrupts chondrocyte function in the GP, causing abnormal bone structure and growth failure. In conditions of insufficient PTHrP signaling, such as some forms of skeletal dysplasia, PTHrP analogs or PTH therapies could theoretically restore normal growth plate and cartilage function, although clinical applications are still under investigation [88].

Heterozygous missense mutations in the *Ihh* gene have been shown to cause brachydactyly type A-1 in humans, a condition involving shortening or absence of the middle phalanges [89]. Conversely, homozygous mutations in *Ihh* cause Acrocapitofemoral Dysplasia, an autosomal recessive disorder characterized by a postnatal-onset disproportionate short stature, relatively large head, narrow thorax, lumbar lordosis, short limbs, and brachydactyly [90]. To date, it seems that the lack of appropriate human-relevant models to accurately represent these chondrodysplasias has hampered the identification of clinically effective treatments.

Disruptions in TGF-β and BMP signaling can stem from genetic mutations, receptor dysfunction, or imbalances in ligand expression. Altered TGF-β signaling can lead to accelerated chondrocyte hypertrophy and calcification, often causing premature cartilage ossification. This is evident in disorders like fibrodysplasia ossificans progressiva (FOP), where excessive BMP signaling drives the ossification of soft tissues, resulting in extra-skeletal bone formation [91]. In conditions where BMP signaling is insufficient or disrupted, such as osteoarthritis (OA), in which osteoblasts receive inadequate signals to sustain bone integrity at joint surfaces, contributing to joint instability and weakened bone structure. TGF-β, typically expressed at high levels in healthy cartilage, is markedly reduced in OA cartilage, leading to compromised chondrogenesis, accelerated cartilage degradation, and osteophyte formation [92]. Additionally, research by Blaney et al. demonstrated that blocking TGF-β in aged cartilage increases susceptibility to damage, underscoring TGF-β’s essential role in supporting cartilage integrity and repair [93]. Osteogenesis imperfecta (OI) is often caused by mutations in the COL1A1 or COL1A2 genes affecting collagen production and has been linked to the dysregulation of TGF-β signaling in cartilage cells. In OI, altered TGF-β activity can enhance chondrocyte hypertrophy while inhibiting osteoblast differentiation, leading to compromised bone formation and increased fragility. HCs secrete factors that can create an inhibitory environment for osteoblasts, such as increased levels of cartilage matrix proteins and inflammatory cytokines. This impairment, in the end, results in fragile bones that are prone to fractures. The important role of TGF-β as a new therapy has been demonstrated by Song et al. which sowed that targeting TGF-β improved osteogenesis imperfecta in patients [94].

The GH/IGF-1 axis is crucial for skeletal development, specifically by promoting chondrocyte proliferation and maturation, essential processes for healthy bone growth [95]. In humans, conditions like primary insulin-like growth factor deficiency (PIGFD) manifest as growth failure despite normal or high GH levels, due to reduced IGF-1 levels, highlighting the necessity of IGF-1 in cartilage maintenance [96]. Nutritional deficits affecting IGF-1 can impair GP function and adult height, since IGF-1 has significant local effects on bone growth beyond its systemic endocrine roles. Chronic kidney disease (CKD) disrupts the GH/IGF-1 axis by impairing kidney function, leading to metabolic bone disease (MBD), which reduces bone formation and mineralization and contributes to growth deficits [97]. CKD often induces GH resistance, where tissues fail to respond adequately to GH, regardless of normal or elevated levels [98]. Additionally, CKD disrupts the calcium–phosphate balance, causing secondary hyperparathyroidism, which impacts GH/IGF-1 signaling and overall bone health. Factors such as inflammation, malnutrition, and chronic stress also contribute to growth delay by inhibiting GH/IGF-1 activity. Elevated pro-inflammatory cytokines, including IL-6, TNF-α, and IL-1β, have been shown to impair IGF-1 signaling and chondrocyte function, intensifying growth delays [99]. Chronic inflammation, seen in conditions such as allergies, infections, autoimmune disorders, and environmental exposures, elevates serum levels of IL-6, TNF-α, and IL-1β. These cytokines not only suppress GH/IGF-1 signaling, inhibiting longitudinal growth, but also directly impair GP function by disrupting IGF-1 intracellular signaling [100,101,102]. A recent consensus statement recommends that children with stage 3–5 CKD or on dialysis should be candidates for GH therapy if they have persistent growth failure. The available studies on the impact of GH on adult height suggest that GH improves adult height in short prepubertal and pubertal CKD patients before and after renal transplantation [99,103,104]. While GH therapy may offer some benefits for adults with CKD, it should be considered on a case-by-case basis; however, some studies have suggested that it could potentially improve bone density and remodeling, although its efficacy in adults as well [105,106].

As previously stated, the crosstalk between chondrocytes and endothelial cells is mediated through various signaling pathways and secreted molecules as VEGF. In OA, it has been shown that increased VEGF levels lead to neovascularization of the cartilage, contributing to inflammation and cartilage degradation. Similarly, in rheumatoid arthritis (RA), the inflammatory environment, driven by cytokines such as interleukin-6 (IL-6) and tumor necrosis factor-alpha (TNF-α), further enhances the crosstalk between chondrocytes and endothelial cells, promoting angiogenesis and subsequent joint destruction. Biologic therapies targeting these key molecules have demonstrated efficacy in managing these diseases. One such example is Bevacizumab (Avastin), a monoclonal antibody that specifically binds to VEGF, preventing its interaction with receptors on endothelial cells and thereby inhibiting angiogenesis. Although the clinical outcomes have been mixed—some studies report improvements in disease activity scores while others indicate limited benefits—the use of Bevacizumab in RA remains off label and continues to be the subject of investigation. These anti-VEGF strategies could potentially reduce inflammation and vascularity not only in RA but also in other joint diseases, such as OA [107]. The exploration of VEGF-targeted therapies in osteoarthritis (OA) is still in its early stages; however, these treatments have the potential to play an important role in disease management. Current research is focused on elucidating the implications of angiogenesis in osteoarthritis and examining how interventions in this process can alleviate symptoms and improve joint function. For instance, studies have indicated that inhibiting pathways associated with VEGF can provide therapeutic benefits in OA by mitigating pain and promoting joint health [108].

Achondroplasia, the most prevalent form of dwarfism, is caused by mutations in the *FGFR3* gene, which encodes the fibroblast growth factor receptor 3 primarily expressed in chondrocytes [109]. This mutation leads to the suppression of VEGF expression, resulting in diminished vascular invasion and nutrient delivery to HCs. Consequently, cell survival is compromised, and the replacement of cartilage with bone is significantly slowed. The chondrocytes located in the hypertrophic zone experience heightened hypoxia due to limited vascularization, triggering apoptotic pathways. This process creates an imbalance, leading to the premature removal of chondrocytes before they can adequately contribute to bone formation, further exacerbating deficiencies in bone growth. Recent research indicates that modulating VEGF levels or enhancing hypoxia-inducible factor 1-alpha (HIF-1α) signaling within growth plate cartilage may partially mitigate the vascular deficiencies associated with achondroplasia. For instance, elevating VEGF expression has shown promising results in animal models, suggesting the potential to improve vascularization, enhance chondrocyte survival, and promote more effective endochondral ossification [110]. In response to these challenges, Ascendis Pharma has developed a novel therapy aimed at improving bone growth by administering a modified form of fibroblast growth factor (FGF) that effectively signals through the *FGFR3* receptor while circumventing the negative regulatory effects of the mutant receptor. Preliminary clinical trials of this therapy have yielded encouraging results, demonstrating significant increases in growth rates and improvements in skeletal development among children with achondroplasia. Notably, a Phase 2 trial reported a marked enhancement in annual growth velocity compared to the placebo in treated patients [111].

Additionally, conditions like X-linked hypophosphatemia (XLH), characterized by defective phosphate regulation, lead to rickets and impaired bone mineralization due to the overactivity of another FGF, specifically FGF23 in osteoblast. The excess FGF23 by osteoblast disrupts IGF signaling in the growth plate, altering chondrocyte hypertrophy and bone growth [97,100,112]. Hereditary hypophosphatemias, are a group of genetic disorders involving renal phosphate loss, characterized by hypophosphatemia, rickets, and normal serum calcium levels. FGF-23 and its growth plate local receptors are increased in hypophosphatemic animal models and has been reported to be responsible for hypophosphatemia and reduced vitamin D levels [100,112], while in vitro analyses of primary murine chondrocytes have demonstrated that phosphate mediates hypertrophic chondrocyte apoptosis by activating the caspase-9-dependent mitochondrial pathway [113]. Children with XLH show an over-activation of both extracellular signal-regulated kinase (ERK) and pERK1/2, leading to an expansion of the hypertrophic chondrocyte layer and a decrease in type I collagen in vitro, as well as an upregulation of the mitogen activated protein kinase (MAPK) signaling pathway in the GP. In this regard, pERK1/2 inhibition activity in Hyp mice relates to a partial recovery of cartilage deformities and skeletal abnormalities [114]. It has been observed that excess FGF23, which is characteristic of rXLH, can influence the signaling of pathways involved in terminal hypertrophy of chondrocytes. Antibody treatments targeting FGF23 have shown promise in alleviating the skeletal complications associated with X-linked hypophosphatemia (XLH) by restoring phosphate homeostasis and improving bone mineralization.

Abnormal Wnt/β-catenin activation in cartilage cells can lead to cartilage degradation and joint inflammation, leading to osteoarthritis (OA). Dysregulation in Wnt signaling can also lead to impaired bone formation, contributing to low bone mass and increased fracture risk in osteoporotic patients. The use of anti-sclerostin antibodies, such as Romosozumab, which were recently approved for osteoporosis treatment [115], directly targets this pathway by inhibiting sclerostin. This inhibition enhances Wnt/β-catenin signaling, promoting osteoblast function, increasing bone formation, and reducing bone resorption, which improves bone density and strength. Anti-sclerostin therapy represents a promising approach, as it could potentially counteract cartilage degeneration and improve joint health in OA if appropriately targeted [116].

Another example can be Sclerosteosis and Van Buchem Disease, which are rare genetic bone disorders characterized by excessive bone formation due to mutations affecting the *SOST* gene [117]. When *SOST* is mutated or its function is diminished, Wnt signaling becomes hyperactive, leading to increased bone density.

One disease that affects a critical transcription factor in the Wnt signaling pathway is the cleidocranial dysplasia (CCD), a rare genetic disorder caused by mutations in the *RUNX2* gene and characterized by the abnormal development of bones and teeth, most notably the skull and clavicles (collarbones). The condition primarily affects osteoblasts impairing the maturation and function of these cells. Additionally, chondrocytes are indirectly affected [118].

Additionally, alterations in the *RANKL* signaling pathway can also cause bone disorders. Excessive *RANKL* activity can lead to abnormal osteoclast formation, disrupting normal GP function and resulting in conditions like osteopetrosis or osteoporosis. Conversely, impaired *RANKL* signaling may delay vascular invasion [119]. It is due to dysregulation of the *RANKL* pathway leading to abnormal bone remodeling, with increased osteoclast activity followed by compensatory osteoblast activity. Characteristics of this disease are as follows: enlarged and deformed bones, pain, and increased risk of fractures. The disorder results in excessive and disorganized bone formation. Miyagawa K et al. reported that high levels of IGF1 in osteoclasts stimulate osteocytes to produce *RANKL*, which promotes abnormal osteoclast activity and pagetic bone lesions formation [120]. Moreover, it has been reported that in patients with rheumatoid arthritis (RA), *RANKL* is highly expressed in synovial tissues and is involved in osteoclast development and thus bone destruction. RA is an inflammatory disorder characterized by progressive joint destruction. Denosumab, a specific antibody to human *RANKL* currently used to treat osteoporosis, efficiently suppressed the progression of bone destruction in patients with RA in a randomized controlled study and is considered a putative therapeutic option for RA. It mimics the action of OPG by binding to *RANKL* and preventing it from activating RANK [121,122]. In conclusion, the disruption of the *RANKL* signaling pathway can lead to significant bone growth disorders characterized by either excessive bone formation or excessive bone resorption. Understanding the mechanisms underlying these disruptions is essential for developing targeted therapies to manage these conditions effectively. Inherited conditions such as pycnodysostosis arise from defects in osteoclast function [123]. These diseases often present as a combination of osteosclerosis and osteolytic lesions due to osteoclast hyperactivity. In such conditions, the primary defect is hyperactivity of osteoclasts. In pycnodysostosis, mutations in *CTSK* lead to a deficiency or dysfunction of cathepsin K, resulting in impaired collagen degradation. Without functional cathepsin K, osteoclasts cannot effectively degrade type I collagen in the bone matrix. These inefficient osteoclasts fail to initiate proper signaling thought RANK-*RANKL*, disrupting the remodeling cycle and resulting in dense but structurally unsound bone. This leads to reduced resorption capacity [124].

The variability of skeletal disorders due to defects in intra- and/or intercellular signaling pathways shows the complex approach of pathologies affecting both bone formation and remodeling. Each pathway, disrupted either through genetic mutations, inflammatory conditions, or metabolic dysregulation, affects the intricate balance between the activity of osteoblasts and osteoclasts, the maturation of chondrocytes, and the overall integrity of the skeleton. Recent breakthroughs using targeted therapies, including inhibitors of *RANKL* and antisclerostin antibodies, bring some very encouraging pathways to rectify some of these imbalances. However, there are still some challenges in developing effective treatments, mainly in cases of rare genetic disorders and conditions with alteration of partially overlapping signaling pathways. Further research focused on elucidating the mechanisms underlying bone cell signaling and crosstalk will allow further fine-tuning of therapeutic strategies in efforts toward more successful outcomes for patients suffering from these multifaceted skeletal diseases.

## 5. Conclusions

In this article, we discuss the complex communications between chondrocytes, osteoblasts, osteoclasts, and endothelial cells that mediate skeletal growth. Cell–cell interactions and related signaling pathways, such as PTH/PTHrP, *Ihh*, TGF-β/BMP, and Wnt/β-catenin, are critical for the development of a skeleton, in which all parts must grow harmoniously through precise pathways such as endochondral ossification. These pathways involved in cell–cell communication that guide growth plate dynamics could serve as promising new avenues of research to improve or correct both primary and non-genetic skeletal disorders.

## Figures and Tables

**Figure 1 biomedicines-12-02662-f001:**
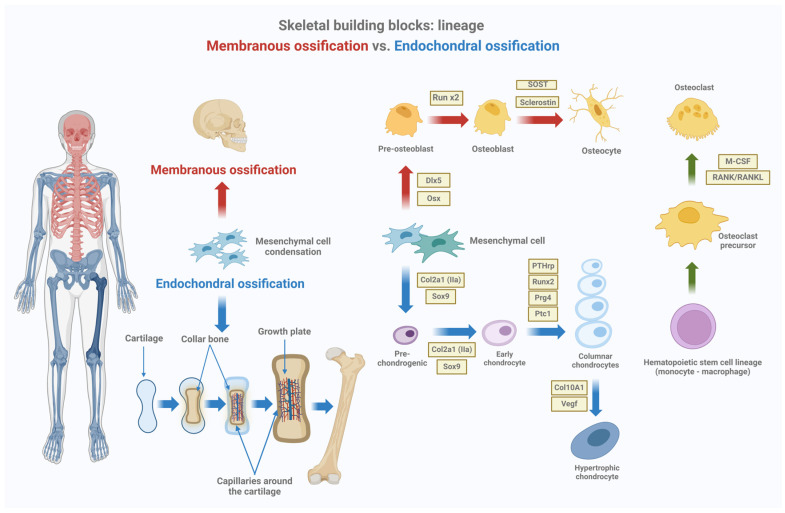
Membranous ossification vs. endochondral ossification. Mesenchymal stem cells (MSCs) present in the bone marrow can differentiate to become either chondrocytes or osteoblasts. Osteoblasts build up bone directly through a process called intramembranous ossification (in red), while chondrocytes proliferate, hypertrophy, and mineralize, and then new bone is deposited onto the cartilaginous matrix through a process called endochondral ossification (in blue). Endochondral ossification forms the bones of the limbs and long bones, while membranous ossification forms the bones of the axial skeleton. In this process, various factors and precursors are involved, from mesenchymal cell differentiation to hypertrophic chondrocyte. the green arrows indicate the differentiation of precursor cells into osteoclasts, which are cells involved in bone resorption. M-CSF and RANK/RANKL induce the differentiation of osteoclast precursors into mature osteoclasts (green arrow). Created using BioRender.com.

**Figure 2 biomedicines-12-02662-f002:**
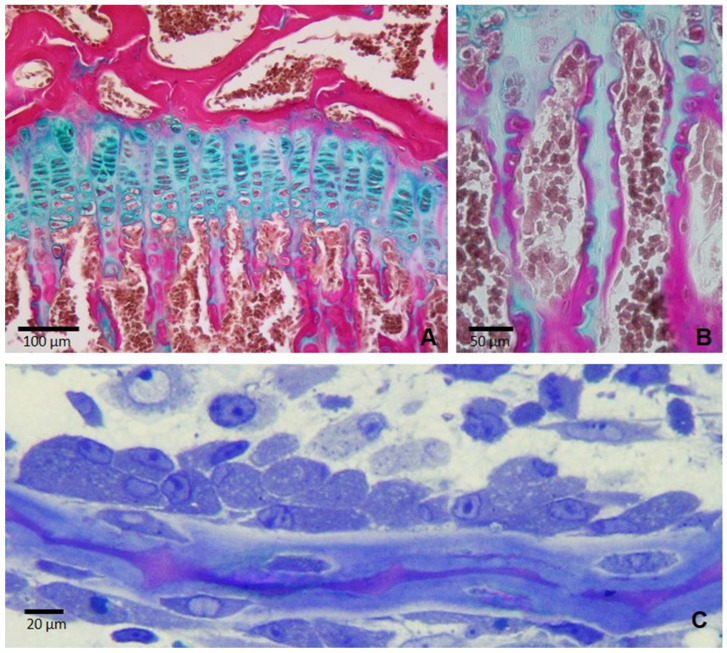
Histological sections of the tibial epiphysis of a rat of 35 days of age. (**A**) A paraffin section stained with Alcian blue/acid fuchsin showing the cartilage of the growth plate (stained in blue) and the bone tissue (stained in fuchsia) in a typical endochondral ossification process where the cartilage is progressively replaced by bone. (**B**) A magnification of figure (**A**) showing that longitudinal septa of the growth plate serve as a scaffold upon which osteoblasts deposit mineralized osseous matrix. (**C**) A semi-thin section of a rat tibia showing a group of osteoblasts secreting bone matrix on a bone trabecula in a representative membranous ossification process.

**Figure 3 biomedicines-12-02662-f003:**
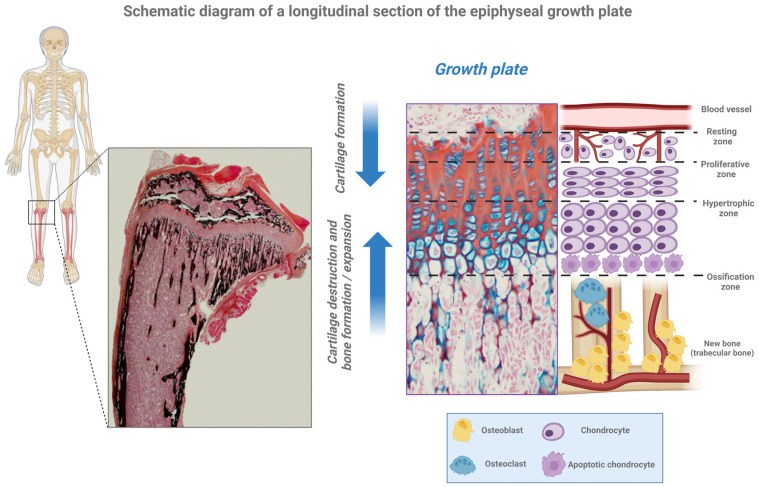
Schematic diagram of a longitudinal section of the epiphyseal growth plate. The growth plate is a cartilage-like structure situated between the metaphysis and the diaphysis of all long bones. It consists of hyaline cartilage. The growth plate is histologically made up of 4 zones. The epiphysis is above the reserve zone, followed by the proliferative and prehypertrophic zones; finally, the metaphysis is below, called the hypertrophic zone. Created using BioRender.com.

**Figure 4 biomedicines-12-02662-f004:**
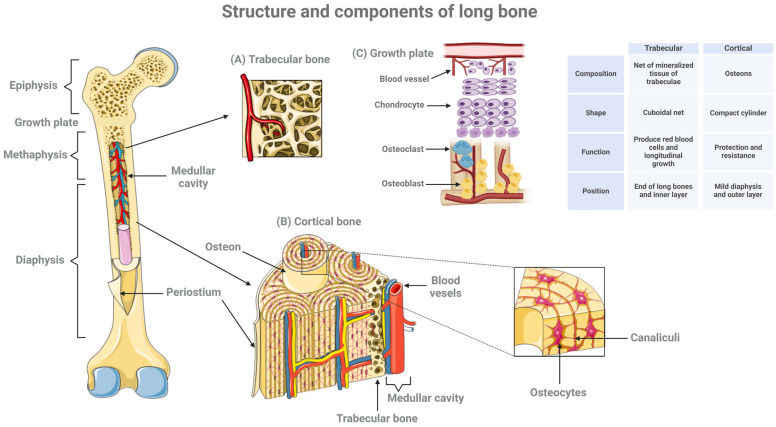
Structure and components of long bones. Long bones consist of a long shaft (the diaphysis) plus two articular (joint) surfaces, called epiphyses. They are composed mostly of compact bone, but they also contain spongy or trabecular bone and marrow in the hollow center (the medullary cavity). The diagram shows the main structural features of bone as well as a magnified view showing some of the finer details of trabecular (**A**) and cortical bones (**B**) and the growth plate (**C**). Created using BioRender.com.

**Figure 5 biomedicines-12-02662-f005:**
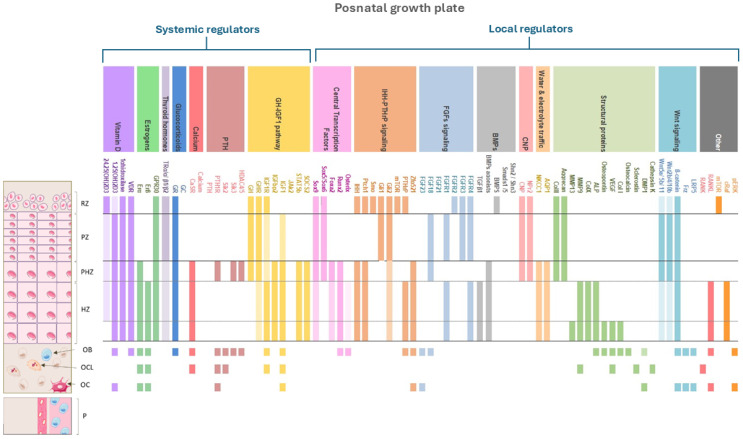
Molecular expression of the main factors involved in the dynamics of the postnatal growth plate. Solid colored bars indicate the protein’s expression in that specific region. Bars with only oblique stripes (no solid color) indicate areas where the protein is not expressed but performs its function through signaling or other indirect means. Bars with both color and oblique stripes represent regions where the protein is actively expressed and the cells in that zone also possess receptors for that protein, allowing direct autocrine or paracrine effects. RZ: resting zone; PZ: proliferative zone; PHZ: prehypertrophic zone; HZ: hypertrophic zone; THZ: terminal hypertrophic zone; OB: osteoblasts; OCL: osteoclasts; OC: osteocytes; 24,25(OH)2D3: 24,25-dihydroxyvitamin D3; 1,25(OH)2D3: 1,25-dihydroxyvitamin D3; VDR: vitamin D receptor; 1α-hydroxylase: 1-alpha hydroxylase; ERα: estrogen receptor alpha; ERβ: estrogen receptor beta; GPR30: G-protein coupled receptor 30; TRα/β1/β2: thyroid hormone receptors α and β1/2; GR: glucocorticoid receptor; GC: glucocorticoids, CaSR: calcium-sensing receptor; PTH: parathyroid hormone; *PTH1R*: parathyroid hormone 1 receptor; Sik2/3: salt-inducible kinase 2/3; HDAC4/5: histone deacetylase 4/5; GH: growth hormone, GHR: growth hormone receptor; IGF1R: insulin-like growth factor 1 receptor; IGFbp2: insulin-like growth factor binding protein 2; IGF1: insulin-like growth factor 1; JAK2: Janus kinase 2; STAT5b: signal transducer and activator of transcription 5b; SOCS2: suppressor of cytokine signaling 2; Sox: SRY-box transcription factor; Foxa2: forkhead box A2; *RUNX2*: runt-related transcription factor; Osx: Osterix or Sp7; *Ihh*: Indian hedgehog; Ptch1: patched-1; Smo: smoothened; Gli1/Gli2: GLI family zinc finger 1 and 2; mTOR: mechanistic target of rapamycin; PTHrP: parathyroid hormone-related protein; Zfp521: zinc finger protein 521; FGF: fibroblast growth factor; FGFR1/2/3/4: fibroblast growth factor receptors 1, 2, 3, and 4; TGFβ1: transforming growth factor beta 1; BMP: bone morphogenetic protein; Smad1/5: SMAD family members 1 and 5; Shn2/Shn3: Schnurri-2 and Schnurri-3; CNP: C-type natriuretic peptide; NPr2: natriuretic peptide receptor 2; NKCC1: Na-K-Cl cotransporter 1; AQP1: aquaporin 1; Col II: type II collagen; ACAN: cartilage-specific proteoglycan core protein; MMP9/13: matrix metalloproteinase 9/13; Col X: type X collagen; ALP: alkaline phosphatase, OPN: osteopontin; VEGF: vascular endothelial growth factor; Col I: type I collagen; BGLAP: bone gamma-carboxyglutamate protein or osteocalcin; DMP1: dentin matrix protein 1; *CTSK*: cathepsin K; Wnt: Wnt family proteins; Frz: frizzled receptors; LRP5: low-density lipoprotein receptor-related protein 5; RANK: receptor activator of nuclear factor kappa-B; *RANKL*: RANK ligand; cRaf: Raf kinase; pERK: phosphorylated extracellular signal-regulated kinase.

**Figure 6 biomedicines-12-02662-f006:**
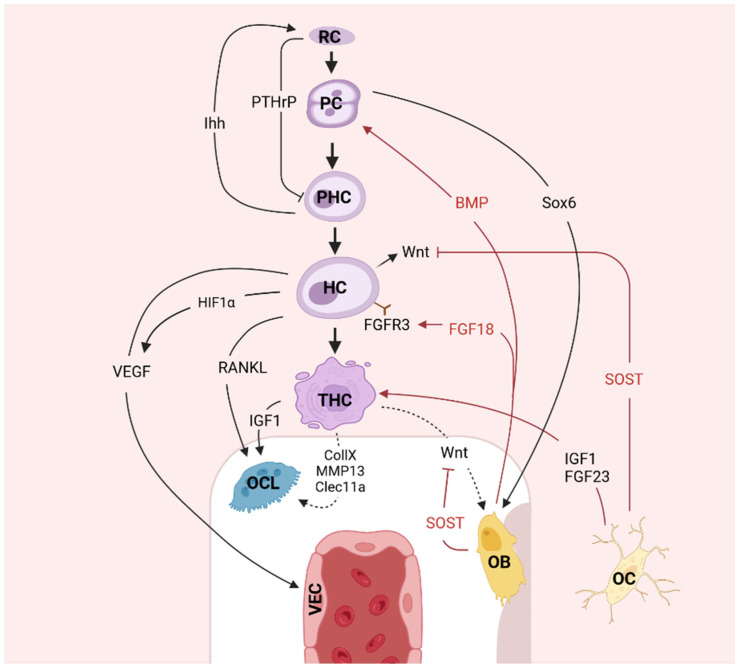
Schematic representation of cellular interactions in the growth plate environment. The diagram illustrates the different stages of chondrocytes from resting chondrocytes (RCs) through proliferative chondrocytes (PCs), prehypertrophic chondrocytes (PHCs), hypertrophic chondrocytes (HCs), and terminal hypertrophic chondrocytes (THCs). Key signaling pathways and molecular mediators influencing cell differentiation, including *parathyroid hormone-related protein* (PTHrP), *Indian hedgehog* (*Ihh*), *vascular endothelial growth factor* (VEGF), *insulin-like growth Factor 1* (IGF1), *RANK Ligand* (*RANKL*), *Bone Morphogenetic Proteins* (BMP), and *Fibroblast Growth factor 18* (FGF18), are shown with black and red arrows. Interactions between chondrocytes and surrounding cells, including osteoblasts (OBs), osteoclasts (OCs), and vascular endothelial cells (VECs), highlight factors like IGF1, FGF23, and Wnt signaling in growth plate regulation. Feedback inhibition pathways are also indicated, such as SOST.

**Table 1 biomedicines-12-02662-t001:** Cell Type markers. The markers listed in this table are key proteins or molecules that are commonly used to identify and characterize the stages of differentiation and functional states of each cell type involved in bone growth, remodeling, and repair. Sox 9: sex-determining region Y (SRY) box 9; Col2a1: collagen II; MMP13: matrix metalloprotein 13; Col10a1: collagen X; *RUNX2*: runt-related transcription factor 2; OCN: osteocalcin; MEPE: matrix extracellular phosphoglycoprotein; *PHEX*: phosphate-regulating gene with homologies to endopeptidases on the X chromosome; Osx: osterix; Col I: collagen 1; ALP: alkaline phosphatase; *PTH1R*: parathyroid hormone 1 receptor; RANK: receptor activator of nuclear factor kappa B; TRAF6: tumor necrosis factor receptor-associated factor 6; *CTSK*: cathepsin K; DMP1: dentin matrix acidic phosphoprotein 1; OPN: osteopontin; BSP: bone sialoprotein; TRAP: tartrate-resistant acid phosphatase; OPG: osteoprotegerin; FGF23: fibroblast growth factor 23; SOST: sclerostin; VEGFR-2: vascular endothelial growth factor receptor 2.

Cell Type	Role in Bone Growth and Remodeling	Key Markers
**Chondrocytes**	Primary cells in growth plate; produce and maintain cartilage matrix, undergo proliferation, hypertrophy, and apoptosis	SOX 9, COL2A1, and ACAN (early development); Col10a1, *RUNX2*, and MMP13 (hypertrophy and bone formation)
**Osteoblasts**	Bone-forming cells derived from MSCs; synthesize organic bone matrix and facilitate mineralization	*RUNX2*, Osx, Col1, ALPL, OCN, PHOSPHO-1, *PTH1R*, MEPE, DMP1, *PHEX*, SOST, and FGF23
**Osteoclasts**	Multinucleated cells from hematopoietic stem cells; responsible for bone resorption	RANK, TRAF6, *CTSK*, and TRAP
**Osteocytes**	Mature osteoblasts embedded in bone matrix; act as mechanosensors and regulate bone formation and resorption	OPN, DMP1, BSP, OPG, SOST, FGF23, and MEPE
**Vascular cells**	Support vascular invasion in hypertrophic zone, deliver nutrients and osteoprogenitor cells, and are necessary for ossification	CD34 and CD31 (endothelial cells); CD133, CXCR4, and VEGFR-2 (endothelial progenitor cells)

**Table 2 biomedicines-12-02662-t002:** Bone disorders: description of various bone diseases caused by mutations in specific genes affecting cartilage and bone main signaling pathways. Each disorder includes its genetic cause, clinical manifestations, and the available treatment options. CKD-MBD: chronic kidney disease–mineral bone disorder; XLH: X-linked hypophosphatemia; OA: osteoarthritis; OI: osteogenesis imperfecta-, RA: rheumatoid arthritis; FOP: fibrodysplasia ossificans progressiva; PIGFD: primary insulin-like growth factor deficiency; CKD-MBD: chronic kidney disease–mineral bone disorder; CCD: cleidocranial dysplasia; NSAIDs: non-steroidal anti-inflammatory drugs. * Experimental practical application.

Involved Pathway	Disease	Genetic Origin	Clinical Manifestations	Authorized Therapy
**PTH/PTHrP**	Blomstrand Chondro-osteodystrophy	*PTH1R* gene	Prenatal lethal disorder Shortened limbs Premature ossification of bones	No
	Jansen Metaphyseal Chondrodysplasia		Abnormal bone growth Short stature Hypercalcemia Metaphyseal widening	Calcium management
** *Ihh* **	Brachydactyly Type A-1	*Ihh* gene	Shortened/absent middle phalanges	No
	Acrocapitofemoral Dysplasia		Short stature Brachydactyly Narrow thorax Short limbs	No
**FGF**	Achondroplasia	*FGFR3* gene	Shortened limbs Large head Spinal stenosis Hypoplasia of the foramen magnum	Anti-VEGF Anti-*FGFR3* Orthopedic surgery
	XLH	*PHEX* gene	Rickets Bone pain Skeletal deformities Impaired bone mineralization	Phosphate supplementation Active vitamin D analogues Anti-FGF23 * Orthopedic surgery
**VEGF**	OA	*VEGFA* gene [SNP]	Joint pain Cartilage loss Decreased joint function	No Anti-VEGF *
	RA	*VEGFA* gene [SNPs] [86]	Tenderness Nocturnal pain Limited joint motion	Anti-VEGF
**BMP/TGF**β	FOP	*ACVR1* gene	Progressive ossification of soft tissues Restricted movement Joint pain	No Anti- BMP *
	OA	*ALDH1A2*, *COLGALT2*, *GDF5*, *MGP*, *NCOA3* *PLEC*, *RUNX2*, *RWDD2B*, *TGFB1*, and *WNT9A* genes [Methylation: *SUPT3H*, *NCOA3*, and *DOT1L* genes]	Joint pain Cartilage loss Decreased joint function	Anti-TGFβ *
	OI	*COL1A2* gene	Fragile bones Fractures	Anti-TGFβ
**GH/IGF-1**	PIGFD	*IGF1* gene	Growth failure Delayed bone growth	IGF-1 supplementation
	CKD-MBD	Secondary to primary or acquired disorders	Bone formation deficits Growth retardation Secondary hyperparathyroidism	Phosphate control GH therapy
**Wnt/β-catenin**	CCD	*RUNX2* gene	Underdeveloped clavicles Dental abnormalities Delayed bone ossification	No Orthopedic surgery
	Sclerosteosis and Van Buchem Disease	*SOST* gene	Excessive bone growth Thickened skull Neural complications	No Anti-sclerostin *
	OA	*ALDH1A2*, *COLGALT2*, *GDF5*, *MGP*, *NCOA3* *PLEC*, *RUNX2*, *RWDD2B*, *TGFB1*, and *WNT9A* genes [Epigenetic association: *SUPT3H*, *NCOA3*, and *DOT1L* genes]	Joint pain Cartilage loss Decreased joint function	NSAIDs Anti-sclerostin *
**RANK/*RANKL*/OPG**	Paget’s Disease of Bone	*RANKL* gene	Pain Enlarged and deformed bones Increased risk of fractures	Bisphosphonates Calcitonin *RANKL* inhibitors
	Osteopetrosis	*TCIRG1*, *CLCN7*, *OSTM1*, *PLEKHM1*, *SNX10*, *TNFSF11*, and *TNFRSF11A* genes	Increased bone density Brittle bones Nerve compression issues	Bone marrow transplant
	Pycnodysostosis	*CTSK* gene	Osteosclerosis Fractures Short stature Increased bone density	GH therapy Orthopedic surgery

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
