# Peer review of "The Crosstalk Between Cartilage and Bone in Skeletal Growth"

_biomedicines, 2024, doi:10.3390/biomedicines12122662_

Round 1

Reviewer 1 Report

Comments and Suggestions for Authors

The manuscript titled "How Cartilage and Bone Communicate to Regulate Bone Growth" provides a comprehensive discussion on the physiological process of longitudinal growth, the consequences of disrupted interactions on growth, and clinical conditions associated with growth retardation. While the article is quite systematic, the authors should address the following points to enhance the manuscript:

1. In section 2.4, the authors elaborate on the coordinated interaction between cartilage and bone cells, focusing on the role of hypertrophic chondrocytes in osteoblast function. However, the reverse interaction, from osteogenic lineage cells to hypertrophic chondrocytes, is not clearly explained. Although the development and growth of hypertrophic chondrocytes are discussed in detail, the interactive dynamics between these cells require further exploration.

2. In section 3, while it is evident that abnormalities in chondrocytes can lead to growth disorders, the purpose of this section is somewhat unclear, and the disorders mentioned do not seem closely related to the interactions discussed. The authors should carefully reconsider the objective and examples presented in this section.

3. In section 4, a more in-depth discussion on the factors influencing the development of chondrocytes and osteogenic lineage cells is needed. Additionally, the imbalances in interactions between cartilage and bone within the context of these diseases should be more thoroughly examined.

4. Formatting and grammatical errors, such as those found on lines 31, 44, 88, etc., should be meticulously reviewed and corrected.

5. The formatting of the references should be scrutinized and confirmed for consistency with the journal's guidelines.

Author Response

Authors’ Response to Referees’ Comments

We thank the referees for their critical reading of our manuscript now entitled “The crosstalk between cartilage and bone in skeletal growth” (formerly entitled “How cartilage and bone talk to each other to allow bone grow?”).

We appreciate the constructive comments and suggestions of the referees, which helped to improve the quality of the manuscript. The manuscript is revised according to all the comments from the referees. Our point-by-point responses to all the comments are provided below.

Referee #1

The manuscript titled "How Cartilage and Bone Communicate to Regulate Bone Growth" provides a comprehensive discussion on the physiological process of longitudinal growth, the consequences of disrupted interactions on growth, and clinical conditions associated with growth retardation. While the article is quite systematic, the authors should address the following points to enhance the manuscript:

Response: We thank the referee for his/her critical reading of the manuscript

  1. In section 2.4, the authors elaborate on the coordinated interaction between cartilage and bone cells, focusing on the role of hypertrophic chondrocytes in osteoblast function. However, the reverse interaction, from osteogenic lineage cells to hypertrophic chondrocytes, is not clearly explained. Although the development and growth of hypertrophic chondrocytes are discussed in detail, the interactive dynamics between these cells require further exploration.

Response: Following the reviewer’s suggestion, we substantially reorganized the paper and developed a section of Cell-Cell Interactions (3.2) including Chondrocytes-Endothelial cells (3.2.1, lines 510-554), Chondrocytes-Osteoblast and cell trans-differentiation (3.2.2, lines 558-632), Chondrocyte-Osteoclast (3.2.3, lines 636-655), Chondrocytes-Osteocytes (3.2.4, lines 659-677), Osteoblasts-Osteoclasts (3.2.5, lines 681-721) and Osteoblasts-Osteocites (3.2.6, lines 725-773)

  1. In section 3, while it is evident that abnormalities in chondrocytes can lead to growth disorders, the purpose of this section is somewhat unclear, and the disorders mentioned do not seem closely related to the interactions discussed. The authors should carefully reconsider the objective and examples presented in this section.

Response: Following the reviewer’s comments, we substantially restructured this section in order to make its purpose more clear.

  1. In section 4, a more in-depth discussion on the factors influencing the development of chondrocytes and osteogenic lineage cells is needed. Additionally, the imbalances in interactions between cartilage and bone within the context of these diseases should be more thoroughly examined.

Response: Following the reviewer’s comments, we did a more in-depth discussion on the factors influencing the development of chondrocytes and osteogenic lineage cells. We have included an extensive description of various bone diseases caused by mutations in specific genes affecting cartilage and bone and added a table (Table 2) to provide a more comprehensive perspective on the range of affected signaling pathways and their clinical manifestations.

  1. Formatting and grammatical errors, such as those found on lines 31, 44, 88, etc., should be meticulously reviewed and corrected.

Response: Formatting and grammatical errors have been reviewed and corrected.

  1. The formatting of the references should be scrutinized and confirmed for consistency with the journal's guidelines.

Response: References have been reviewed and corrected according to the journal's guidelines.

Reviewer 2 Report

Comments and Suggestions for Authors

The manuscript is a review that attempts to consolidate the current understanding of cellular and molecular mechanisms underlying the communication between the growth plate (GP), bone, and surrounding tissues. The authors have set out to shed light on how growth plate alterations, especially in renal disease, impact growth by causing failures in longitudinal bone development.

 The authors aim to clarify the interaction between chondrocytes and various cells that participate in bone development. The review states, “Since there is poor information about communication between chondrocytes and other cell types in developing bones, this review examines the current knowledge of how interactions between chondrocytes and bone-forming cells modulate bone growth.” However, in order to ensure relevance, the authors could consider revising Sections 2.1 and 2.2 under “Physiological Process of Longitudinal Growth”. These sections, while informative, include fundamental information that may excessively lengthen the manuscript with readily available content. To enhance the manuscript's contribution, the authors should focus on novel insights and delete sections that are widely covered in existing literature.

The authors present a mechanism by which growth hormone (GH) interacts with the growth plate through GH receptors (GHRs) located on chondrocytes and indirectly through systemic and local stimulation of IGF-1. A question arises regarding whether the signaling pathway described for GH/GHR interactions is a universal mechanism applicable across all GP zones, or if its role varies in different physiological conditions affecting growth plates in pathologies like chronic inflammation or renal disease.

The manuscript also describes an in vitro model using cultured fetal rat metatarsal bones that suggests TNF-α and IL-1β can act synergistically on growth plate chondrocytes to inhibit longitudinal growth by decreasing chondrocyte proliferation and increasing apoptosis. The authors should clarify the specific roles and conditions under which these cytokines act to alter the GH/IGF-1 axis in the growth plate, and whether this pathway could be influenced by microRNAs in chronic inflammatory diseases.

In the case of hereditary hypophosphatemic rickets, the manuscript discusses that excess fibroblast growth factor 23 (FGF23), characteristic of rXLH, can disrupt pathways involved in terminal hypertrophy of chondrocytes. Given this, it would be pertinent to further explore if FGF23’s interactions with these pathways have differential impacts on bone growth depending on the expression of IGF1 in these regions.

The final paragraph of Section 4 is repeated in the conclusion. The conclusion itself seems to suggest that the review does not contribute novel insights, stating that significant knowledge gaps remain. This could be misleading, given the title of the manuscript, which implies a thorough synthesis of the current state of knowledge. The authors should revise the conclusion to highlight the unique contributions of this review, emphasizing what new perspectives it brings to the field, and addressing how these insights may inform future research or clinical applications.

Author Response

Authors’ Response to Referees’ Comments

We thank the referees for their critical reading of our manuscript now entitled “The crosstalk between cartilage and bone in skeletal growth” (formerly entitled “How cartilage and bone talk to each other to allow bone grow?”).

We appreciate the constructive comments and suggestions of the referees, which helped to improve the quality of the manuscript. The manuscript is revised according to all the comments from the referees. Our point-by-point responses to all the comments are provided below.

1. The manuscript is a review that attempts to consolidate the current understanding of cellular and molecular mechanisms underlying the communication between the growth plate (GP), bone, and surrounding tissues. The authors have set out to shed light on how growth plate alterations, especially in renal disease, impact growth by causing failures in longitudinal bone development.

Response: We thank the referee for his/her critical reading of the manuscript.

2. The authors aim to clarify the interaction between chondrocytes and various cells that participate in bone development. The review states, “Since there is poor information about communication between chondrocytes and other cell types in developing bones, this review examines the current knowledge of how interactions between chondrocytes and bone-forming cells modulate bone growth.” However, in order to ensure relevance, the authors could consider revising Sections 2.1 and 2.2 under “Physiological Process of Longitudinal Growth”. These sections, while informative, include fundamental information that may excessively lengthen the manuscript with readily available content. To enhance the manuscript's contribution, the authors should focus on novel insights and delete sections that are widely covered in existing literature.

Response: Following the reviewer’s comments, the text of these sections has been extensively rewritten. We have maintained only indispensable information to contextualize the new contributions

3. The authors present a mechanism by which growth hormone (GH) interacts with the growth plate through GH receptors (GHRs) located on chondrocytes and indirectly through systemic and local stimulation of IGF-1. A question arises regarding whether the signaling pathway described for GH/GHR interactions is a universal mechanism applicable across all GP zones, or if its role varies in different physiological conditions affecting growth plates in pathologies like chronic inflammation or renal disease.

Response: we have clarified in the text that the signaling pathway described for GH/GHR interactions is a universal mechanism applicable across all GP zones and that chronic kidney disease disrupts the GH/IGF-1 axis leaving to alterations in chondrocyte proliferation and maturation and this results in a pathological affectation of bone growth

4. The manuscript also describes an in vitro model using cultured fetal rat metatarsal bones that suggests TNF-α and IL-1β can act synergistically on growth plate chondrocytes to inhibit longitudinal growth by decreasing chondrocyte proliferation and increasing apoptosis. The authors should clarify the specific roles and conditions under which these cytokines act to alter the GH/IGF-1 axis in the growth plate, and whether this pathway could be influenced by microRNAs in chronic inflammatory diseases.

Response: we have clarified in the text (starting in line 860) that elevated pro-inflammatory cytokines, including, TNF-α, and IL-1β, have been shown to impair IGF-1 signaling and chondrocyte function, intensifying growth delays. We commented some works reporting that chronic inflammation, seen in conditions such as allergies, infections, autoimmune disorders, and environmental exposures, elevates serum levels of IL-6, TNF-α, and IL-1β and that these cytokines not only suppress GH/IGF-1 signaling, inhibiting longitudinal growth, but also directly impair GP function by disrupting IGF-1 intracellular signaling.

5. In the case of hereditary hypophosphatemic rickets, the manuscript discusses that excess fibroblast growth factor 23 (FGF23), characteristic of rXLH, can disrupt pathways involved in terminal hypertrophy of chondrocytes. Given this, it would be pertinent to further explore if FGF23’s interactions with these pathways have differential impacts on bone growth depending on the expression of IGF1 in these regions.

Response: we have included in the text (starting in line 923) three works reporting that the excess FGF23 by osteoblast disrupts IGF signaling in the growth plate, altering chondrocyte hypertrophy and bone growth.

6. The final paragraph of Section 4 is repeated in the conclusion. The conclusion itself seems to suggest that the review does not contribute novel insights, stating that significant knowledge gaps remain. This could be misleading, given the title of the manuscript, which implies a thorough synthesis of the current state of knowledge. The authors should revise the conclusion to highlight the unique contributions of this review, emphasizing what new perspectives it brings to the field, and addressing how these insights may inform future research or clinical applications.

Response: Following the reviewer’s comments, the text of the conclusion has been completely rewritten. We focused that the complex communications between chondrocytes, osteoblasts, osteoclasts, and endothelial cells are critical for the development of the skeleton.

Reviewer 3 Report

Comments and Suggestions for Authors

The manuscript reviews the crosstalk between chondrocytes and bone-forming cells that regulate endochondral ossification. While the manuscript addresses an important topic in bone biology, it requires significant revisions to improve its clarity, structure, and depth. A more detailed discussion of the molecular mechanisms underlying cell-cell communication, along with an increased focus on pathophysiological backgrounds, would enhance the overall impact of the review. Improvements in English language and scientific correctness are also needed.

The title "How cartilage and bone talk to each other to allow bone grow?" is informal and grammatically incorrect. I suggest a more precise title such as "Communication Between Cartilage and Bone Cells in the Regulation of Bone Growth."

The phrase in the abstract, "Most of our bone are formed through a process called endochondral ossification," is unclear and inaccurately phrased. Could the authors revise this to reflect that endochondral ossification forms most bones, while others, such as flat bones, form through intramembranous ossification?

The manuscript discusses the importance of cross-talk between chondrocytes and other cells but lacks sufficient mechanistic detail. Could the authors elaborate on specific signaling pathways or molecular interactions that mediate this cross-talk?

There should be a more detailed discussion of how these signaling mechanisms facilitate interactions between chondrocytes and osteoblasts, osteoclasts, osteocytes, and vascular cells during bone formation.

The manuscript appears disjointed in its current form. I suggest restructuring the review to include clear sections on each type of cell-cell interaction (e.g., chondrocyte-osteoblast, chondrocyte-vascular cells) and the specific signaling pathways involved in these interactions. Additionally, the manuscript should provide a more focused discussion on crosstalk at different stages of development, including early and late stages of osteoarthritis (OA), which is briefly mentioned but not sufficiently elaborated.

The manuscript would benefit from including a detailed description of the factors involved in the crosstalk between chondrocytes and bone-forming cells.

Could the authors also explore whether any therapeutic strategies have been developed based on modulating these cell-cell interactions?

Comments on the Quality of English Language

English needs to be improved

Author Response

Authors’ Response to Referees’ Comments

We thank the referees for their critical reading of our manuscript now entitled “The crosstalk between cartilage and bone in skeletal growth” (formerly entitled “How cartilage and bone talk to each other to allow bone grow?”).

We appreciate the constructive comments and suggestions of the referees, which helped to improve the quality of the manuscript. The manuscript is revised according to all the comments from the referees. Our point-by-point responses to all the comments are provided below.

1. The manuscript reviews the crosstalk between chondrocytes and bone-forming cells that regulate endochondral ossification. While the manuscript addresses an important topic in bone biology, it requires significant revisions to improve its clarity, structure, and depth. A more detailed discussion of the molecular mechanisms underlying cell-cell communication, along with an increased focus on pathophysiological backgrounds, would enhance the overall impact of the review.

Response: Following the reviewer’s suggestion, we substantially reorganized the paper and deepened in the molecular mechanisms underlying cell-cell communication. Furthermore, we have included a section of disorders of bone growth with clinical Implications and future perspectives (section 4, lines from 775 to 1009)

2. Improvements in English language and scientific correctness are also needed.

Response: Formatting and grammatical errors have been reviewed and corrected.

3. The title "How cartilage and bone talk to each other to allow bone grow?" is informal and grammatically incorrect. I suggest a more precise title such as "Communication Between Cartilage and Bone Cells in the Regulation of Bone Growth."

Response: We have taken this suggestion into account and changed the title to “The crosstalk between cartilage and bone in skeletal growth

4. The phrase in the abstract, "Most of our bone are formed through a process called endochondral ossification," is unclear and inaccurately phrased. Could the authors revise this to reflect that endochondral ossification forms most bones, while others, such as flat bones, form through intramembranous ossification?

Response: We have taken this suggestion into account and specified in the abstract that while the flat bones of the face, most of the cranial bones, and the clavicles are formed directly from sheets of undifferentiated mesenchymal cells, most bones in the human body are first formed as cartilage templates

5. The manuscript discusses the importance of cross-talk between chondrocytes and other cells but lacks sufficient mechanistic detail. Could the authors elaborate on specific signaling pathways or molecular interactions that mediate this cross-talk?

Response: Following the reviewer’s suggestion, we substantially restructured the manuscript including a section (section 3) about the signaling pathways and cell-cell Interactions in bone growth. Furthermore, we included an additional figure (number 6) with a schematic representation of cellular interactions in the growth plate environment.

6. There should be a more detailed discussion of how these signaling mechanisms facilitate interactions between chondrocytes and osteoblasts, osteoclasts, osteocytes, and vascular cells during bone formation.

Response: We have taken this suggestion into account and substantially restructured the manuscript and included a section of Cell-Cell Interactions (3.2) including Chondrocytes-Endothelial cells (3.2.1, lines 510-554), Chondrocytes-Osteoblast and cell trans-differentiation (3.2.2, lines 558-632), Chondrocyte-Osteoclast (3.2.3, lines 636-655), Chondrocytes-Osteocytes (3.2.4, lines 659-677), Osteoblasts-Osteoclasts (3.2.5, lines 681-721) and Osteoblasts-Osteocites (3.2.6, lines 725-773).

7. The manuscript appears disjointed in its current form. I suggest restructuring the review to include clear sections on each type of cell-cell interaction (e.g., chondrocyte-osteoblast, chondrocyte-vascular cells) and the specific signaling pathways involved in these interactions. Additionally, the manuscript should provide a more focused discussion on crosstalk at different stages of development, including early and late stages of osteoarthritis (OA), which is briefly mentioned but not sufficiently elaborated.

Response: We have taken this suggestion into account and substantially restructured the manuscript and included a sections in which the cell-cell interactions of the different cell types are considered

8. The manuscript would benefit from including a detailed description of the factors involved in the crosstalk between chondrocytes and bone-forming cells.

Response: Following the reviewer’s suggestion, we have a detailed description of the factors involved in the crosstalk between chondrocytes and bone-forming cells (section 3.2. Cell-Cell Interactions, lines 486-773)

9. Could the authors also explore whether any therapeutic strategies have been developed based on modulating these cell-cell interactions?

Response: Following the reviewer’s suggestion, we have included a text (lines 882-896) indicating that biologic therapies targeting some key molecules have demonstrated efficacy like Bevacizumab (Avastin), a monoclonal antibody that specifically binds to VEGF, preventing its interaction with receptors on endothelial cells and thereby inhibiting angiogenesis. It is indicated that some studies report improvements in disease activity scores while others indicate limited benefits, but the use of Bevacizumab in RA remains off-label and continues to be the subject of investigation. Likewise, it is indicated  the exploration of VEGF-targeted therapies in osteoarthritis.

Round 2

Reviewer 1 Report

Comments and Suggestions for Authors

The manuscript has been sufficiently improved, I agree the article to be published in this journal.

Reviewer 2 Report

Comments and Suggestions for Authors

In light of the responses received to my observations, this reviewer has no further comments.

Reviewer 3 Report

Comments and Suggestions for Authors

The authors have made substantial changes in several part of the paper and addressed the reviewers’ comments. This manuscript may be accepted for publication.